# Biological Functions of the DNA Glycosylase NEIL3 and Its Role in Disease Progression Including Cancer

**DOI:** 10.3390/cancers14235722

**Published:** 2022-11-22

**Authors:** Lang Chen, Xuan Huan, Xi-Dan Gao, Wu-Han Yu, Guo-Hui Xiao, Teng-Fei Li, Zhe-Yuan Wang, You-Cheng Zhang

**Affiliations:** 1Laboratory of Hepatic-Biliary-Pancreatic, Department of General Surgery, Second Hospital of Lanzhou University, Lanzhou 730030, China; 2Second Hospital of Lanzhou University, Lanzhou 730030, China

**Keywords:** DNA glycosylase, NEIL3, cancer

## Abstract

**Simple Summary:**

Under the continuous stimulation of oxidizing substances, DNA bases undergo oxidative damage and excision repair alternately at all times. Once this process becomes abnormal, many pathological diseases will appear one after another. NEIL3 is an important DNA glycosylase in the base excision repair pathway and is involved in oxidative DNA base damage repair. In recent years, several studies have confirmed that abnormal expression of NEIL3 is closely associated with cancers and cardiovascular and neurological diseases, and it may be a potential prognostic molecule and therapeutic target. circNEIL3, a circular RNA of exon-encoded origin by NEIL3, also promotes the development of multiple cancers. In this review, we not only describe the structural specificity and functional specificity of NEIL3, but also focus on NEIL3 and circNEIL3 in cancer progression and prognosis. In addition, we also discuss the link between NEIL3 and the progression of cardiovascular and neurological diseases.

**Abstract:**

The accumulation of oxidative DNA base damage can severely disrupt the integrity of the genome and is strongly associated with the development of cancer. DNA glycosylase is the critical enzyme that initiates the base excision repair (BER) pathway, recognizing and excising damaged bases. The Nei endonuclease VIII-like 3 (NEIL3) is an emerging DNA glycosylase essential in maintaining genome stability. With an in-depth study of the structure and function of NEIL3, we found that it has properties related to the process of base damage repair. For example, it not only prefers the base damage of single-stranded DNA (ssDNA), G-quadruplex and DNA interstrand crosslinks (ICLs), but also participates in the maintenance of replication fork stability and telomere integrity. In addition, NEIL3 is strongly associated with the progression of cancers and cardiovascular and neurological diseases, is incredibly significantly overexpressed in cancers, and may become an independent prognostic marker for cancer patients. Interestingly, circNEIL3, a circular RNA of exon-encoded origin by NEIL3, also promotes the development of multiple cancers. In this review, we have summarized the structure and the characteristics of NEIL3 to repair base damage. We have focused on NEIL3 and circNEIL3 in cancer development, progression and prognosis.

## 1. Introduction

Endogenous or exogenous factors such as inflammation, oxidative stress, ionizing radiation, and metabolites can promote the production of reactive oxygen species (ROS) or reactive nitrogen species (RON), including O_2_^•−^, H_2_O_2_, HO^•^, NO and ONOO^−^ [1,2,3]. Their continued accumulation causes oxidative damage to DNA bases, such as 5-hydroxyuracil (5-OHU), 5-hydroxycytosine (5-OHC), thymine glycol (Tg), 8-oxoguanine (8-oxoG), 2,6-diamino-4-hydroxyformamidopyrimidine (FapyG) and FapyA [2,4,5,6]. If these damages are not repaired quickly, they may lead to base mismatches, blocked DNA replication or transcription, disrupting the stability and integrity of the genome, ultimately leading to aging and cancer development.

Base excision repair (BER) is the main pathway responsible for repairing most oxidative DNA base damage; this pathway mainly includes DNA glycosylases, nucleic acid endonucleases, DNA polymerases and DNA ligases, which are highly conserved in bacteria or mammals [7]. DNA glycosylase is the first enzyme that initiates the BER pathway, which explicitly recognizes the damaged base. It then releases the damaged base by cutting the N-glycosyl bond between the deoxyribose and the base, leading to the production of purine-free/pyrimidine-free (AP) sites (glycosylase activity) [8,9]. Monofunctional DNA glycosylases recruit AP nucleic acid endonuclease (APE1) and cleave the phosphodiester bond at the 5’ end of the AP site, which produces a single nucleic acid gap with 3’-OH and 5’-P termini [9]. Moreover, in addition to cutting damaged bases, the bifunctional enzyme has an intrinsic AP cleavage enzyme activity and can cut the DNA backbone at the AP site through a β-elimination or β/δ-elimination mechanism and proceeds to the repair phase of the DNA strand [2,8]. NEIL3 is a member of the DNA glycosylase family whose expression is tightly regulated in time and space, with high expression mainly in cells with high proliferative capacities such as embryos, thymus, spleen, bone marrow and cancer, and plays a vital role in recognizing and cleaving damaged bases [10,11,12]. With an in-depth study of NEIL3 structure and function, we find that it has characteristics associated with the process of base damage repair; for example, it prefers base damage in single-stranded DNA (ssDNA) or G-quadruplex, maintains replication fork stability, preferentially repairs telomeres and unravels DNA interstrand crosslinks (ICLs). In addition, studies in recent years have revealed that NEIL3 and its epitope encoding source, circNEIL3, are overexpressed in various cancers and may act as an oncogene to promote cancer development and progression. This review reviews the latest frontiers in NEIL3 repairs of oxidized DNA base damage. More importantly, we focus on the role and prognostic value of NEIL3 and circNEIL3 in cancer progression, which we hope will assist research into NEIL3 and cancer.

## 2. DNA Glycosylase Family

In an earlier study, researchers first identified two bacterial DNA glycosylases from E. coli: the formamidopyrimidine (Fapy) DNA glycosylase (Fpg) and the nucleic acid endonucleas1e VIII (Nei) [13]. Fpg is mainly responsible for repairing oxidized purines, such as 8-oxoG and FapyG; the Nei protein sequence has significant homology with Fpg, and its specific substrates are oxidized pyrimidines and FapyA [14]. Later, researchers identified five DNA glycosylases with essential roles in mammals: NTH1, OGG1, and nucleic acid endonuclease VIII-like proteins (NEIL1, NEIL2, and NEIL3) [15]. Their distribution and expression patterns differ significantly in human or rodent brains, with NTH1, OGG1, NEIL1 and NEIL2 being widely expressed at all ages. In particular, NEIL1 expression levels in mouse brains gradually increase with age [16], unlike NEIL3, which is expressed only in the stem-cell-rich region of the brain and gradually decreases with age [16].

Reportedly, Fpg/Nei, NTH1, OGG1, and NEIL1/2/3 all possess the general activity of DNA glycosylases (cleaving damaged bases) and AP lyase activity, which are bifunctional [17,18]. Interestingly, in the presence of APE1, human NEIL3 may function mainly as a single functional enzyme, as the vigorous glycosylase and weak AP lyase activity of NEIL3 exhibit discordant properties during the repair process [8,19]. In the mechanism of cleaving the single-stranded phosphate backbone of DNA, Fpg/Nei and NEIL1/2 produce single-nucleotide gaps with 3’-P and 5’-P termini by β,δ-elimination mechanisms, following which they rely on polynucleotide kinase phosphatase (PNKP) activity to modify the 3’-P termini and produce normal 3’-OH termini [19,20,21]. Conversely, NTH1, OGG1 and NEIL3 mainly rely on the β-elimination mechanism to form 3’-αβ-unsaturated aldehyde and 5’-P terminus at the AP site and then recruit APE1 to modify 3’-αβ-unsaturated aldehyde and generate normal 3’-OH terminus [21,22]. Finally, the BER pathway recruits DNA polymerase and DNA ligase to repair ssDNA gaps [19].

DNA glycosylases of mammalian origin may differ in substrate specificity. NTH1 and OGG1 have the general characteristics of members of the helix–hairpin–helix (HhH) superfamily and are direct homologs of the Nth family in *E. coli*. NTH1 and OGG1 are active only on double-stranded DNA, cleaving mainly oxidized pyrimidines and purines [15,21,22]. A study confirmed that 8-oxoG was mainly recognized and excised by OGG1, and when OGG1 was depleted or inhibited, NEIL1 and NEIL2 aggregated at sites of 8-oxoG damage as backup BER enzymes that complemented OGG1 [23]. NEIL1 and NEIL2 remove damaged bases from not only double-stranded DNA (dsDNA) and single-stranded DNA (ssDNA), but also remove 5-OHU damage in DNA bubbles [21,24]. Alternatively, they may be preferentially involved in repairing DNA damage during DNA replication or transcription [15]. For example, NEIL1 binds DNA-replication-associated proteins in the S phase and participates in the repair process of damaged bases before replication, while NEIL2 mainly functions as a standby enzyme [25]. Similarly, NEIL1 preferentially repairs 5-OHU and Tg damage upstream of the replication fork of dsDNA, but it shows lower enzymatic activity at replication fork junctions or damaged bases in ssDNA [9]. NEIL2 interacts with RNA polymerase II during transcription and prefers to repair oxidatively damaged bases in transcribed genes [26]. The NEIL1/2/3 pairs of modified bases have extensive and overlapping characteristics. However, unlike NEIL1 and NEIL2, NEIL3 still has relatively unique characteristics in structure and function, which we will describe in detail in the following sections.

## 3. The Structure of Neil3 Has Unique Characteristics

The human NEIL3 gene is located on chromosome 4q34.3. NEIL3 protein consists of approximately 605 amino acid residues, and its molecular weight is around 68 kDa, almost twice the size of other Fpg/Nei family members [5]. In 2009, Krokeide et al. first expressed and purified the human NEIL3 protein [4]. In contrast to other members of the Fpg/Nei family of proteins, they found that the N-terminal end of the human NEIL3 protein has an integral Fpg/Nei-like core protein structural domain (GD), consisting mainly of the N-terminal domain, a helix-2-turn-helix (H2TH) motif and a zinc finger DNA-binding motif [5,27,28]. Additionally, in the N-terminal structural domain of NEIL3, valine replaces the central catalytic role of proline; it also has been shown that activation of the glycosylase activity of NEIL3 requires the removal of the N-terminal methionine [5]. Additionally, unlike NEIL1/2, NEIL3 protein includes a RANbp-like zinc finger (ZNF) motif and a duplicated GRF-zinc finger (GRF-ZF) motif at the C-terminus, consisting of approximately 323 amino acid residues, which is a unique feature of NEIL3 protein structure [28] (Figure 1). GRF-ZF is a sequence consisting of glycine, arginine and phenylalanine, which is also present in many DNA/RNA-related processing enzymes (like APE2 and human DNA topoisomerase IIIa) besides NEIL3 [4,29]. It was shown that GRF-ZF mainly acts as an ssDNA binding element to assist NEIL3 in repairing damaged bases, which may be one of the mechanisms by which NEIL3 prefers ssDNA [28]. Reportedly, NEIL3 GRF-ZF repeats, but not individual GRF-ZF, showed a high affinity for shorter ssDNA, and they also found that NEIL3 relied on its GRF-ZF repeats to interact with APE1 and inhibit its activity, reducing the proportion of APE1-mediated ssDNA breaks and protecting DNA from damage caused by oxidative stress [29].

## 4. The Characteristics of NEIL3 for Repairing Oxidative DNA Base Damage

### 4.1. NEIL3 Prefers ssDNA-Derived Base Damage

Unlike NEIL1/2, human NEIL3 or mouse NEIL3 (MmuNeil3) has a better preference for base damage in ssDNA, cutting bases on the side of the replication fork [7,9]. Liu et al. reported the first crystal structure of the mouse NEIL3 enzyme: MmuNeil3Δ324, and they found that MmuNeil3Δ324 has a higher intrinsic affinity for ssDNA [7]. In addition, they further experimentally confirmed that this could be caused by two factors [7]: (1) NEIL3 carried two negatively charged residues (Asp-133 and Glu-269) along the profile of the opposite strand, which disrupted the affinity of NEIL3 for dsDNA; (2) NEIL3 lacked four residues, including the two residues (Phe and Arg) of the “gap-filling triplet”, which interacted with the phosphate backbone of the opposite strand in double-stranded DNA and stabilized the opposite strand. The hydantoin lesions are the further oxidation products of 8-oxoG, including spiroiminodihydantoin (Sp) and guanidinohydantoin (Gh) [30,31]. Several studies have confirmed that NEIL3 could not excise 8-oxoG, yet NEIL3 was extremely sensitive to Sp and Gh in the single-chain environment, suggesting that NEIL3 is a crucial enzyme for removing hydantoins [8,32].

### 4.2. NEIL3 Maintains the Stability of the Replication Fork

During DNA replication, DNA at the replication fork undergoes unwinding. It forms a localized ssDNA replication template susceptible to DNA damage and DSB breaks, resulting in replication fork instability and replication stalling; such factors that interfere with DNA replication are collectively referred to as replication stress [33]. Therefore, the organism must repair the replication fork appropriately to maintain genome stability during replication. It has been found that the transcription and translation levels of NEIL3 exhibited strong fluctuating changes in the cell cycle, starting to be induced in the early S phase and peaking in the G2 phase, suggesting that NEIL3 plays a role in the replication process [12,34]. Rad51 is an essential protein in the homologous recombination (HR) pathway and is recruited upstream of the replication fork during replication, participating in biological processes such as maintaining replication fork stability, DSB repair and restarting replication [35]. A study suggested that NEIL3 may activate the HR pathway during DNA replication (S phase) by promoting the binding of Rad51 to replication forks, acting as a stabilizer of replication forks, while NEIL3-deficient cells exhibited stalled replication forks and significantly increased DSB [36]. Replication protein A (RPA) is an ssDNA-binding protein binding to ssDNA that activates downstream ATR/CHK-1 kinases [37]. The latter stabilizes stalled replication forks by initiating a series of downstream DNA repair pathways; however, significant depletion of RPA instead accelerates the onset of replication-associated DSB [37]. As reported, the deficiency of NEIL3 induced massive aggregation of RPA at the replication fork. Although this contributed to the stability of the replication fork, the massive aggregation of RPA at the replication fork depleted RPA and blocked the binding of Rad51 to the nascent DNA strand, ultimately increasing the occurrence of DSB [36]. The above studies illustrate that NEIL3 is involved in stabilizing the replication fork process, and the deficiency of NEIL3 will lead to instability changes in the replication-related genome and inhibit the replication fork process.

### 4.3. NEIL3 Repairs DNA Damage in G-Quadruplex Structures

G-quadruplex is a four-stranded DNA structure enriched mainly in the telomeric and promoter regions of genes [38,39,40]. Under physiological salt solution conditions, it has been shown that every fourth guanine forms guanine quartets through hoogsteen base pairing, and two or more layers of guanine quartets further form stable parallel G-quadruplex structures [38,39,41]. G-quadruplex plays a crucial role in regulating telomere biological functions and gene transcription, as it inhibits DNA replication, transcription and mRNA translation processes, and hinders telomeric DNA lengthening by inhibiting telomerase activity [41,42]. Therefore, in oncology, G-quadruplex is considered a novel anti-cancer therapeutic target, and some small molecules have been shown to exert certain anti-cancer effects by stabilizing the structure of the G-quadruplex [43]. Due to the richness of guanine, the G-quadruplex structure is susceptible to oxidative damage, mainly forming damaged bases such as 8-oxoG, Sp or Gh; these damaged bases will reduce the thermal stability of the G-quadruplex structure, change its folding pattern at the base damage site and form an inverse parallel G-quadruplex structure [40,44]. Therefore, the organism must have DNA repair systems that repair base damage in telomeres or promoter G-quadruplex structures.

The excision activity of NEIL1, NEIL2, mNeil3 (the direct mouse homolog of human NEIL3), NTH1 and OGG1 on base damage in the telomeric G-quadruplex was evaluated [44]. It was found that NEIL1 and mNeil3 effectively removed Sp and Gh damage from G-quadruplex DNA, but none of the five glycosylases (NEIL1, NEIL2, mNeil3, NTH1 and OGG1) showed activity against 8-oxoG damage, even though 8-oxoG was the primary substrate for OGG1 [44]. Interestingly, only mNeil3 showed vigorous glycosylase activity against Tg damage in the telomeric G-quadruplex, with significantly better excision efficiency than other DNA glycosylases, while NEIL1 seemed to prefer Gh [44]. This study suggests that NEIL3 and NEIL1 are crucial in the repair of telomeric G-quadruplex DNA damage and are essential glycosylases for maintaining telomere stability. Another study reported that c-MYC or VEGF promoter sequences formed parallel quadruplex/triplex structures in physiological K^+^ buffer systems [40]. When guanine was replaced with 8-oxoG or Gh, they found that none of the five glycosylases could remove 8-oxoG and Gh [40]. In contrast, the VEGF promoter sequence formed a mixture of parallel and antiparallel quadruplex DNA in a physiological Na^+^ buffer system, and both NEIL1/2/3 glycosylases efficiently removed Gh damage [40]. This study illustrates that NEIL1/2/3 primarily removes hydantoin lesions from promoter quadruplex DNA with inverse parallelism.

### 4.4. NEIL3 Repairs DNA Damage in Telomere Structures

The telomere is a DNA–protein complex covering the end of the chromosome and plays a vital role in maintaining the stability of chromosomal DNA [45]. The human telomere structure consists of four main components: a DNA sequence consisting of tandemly repeated 5’-TTAGGG-3’, Shelterin proteins (TRF1, TRF2, POT1, TPP1, TIN2, RAP1) bound to the chromosome end, CST complex and telomerase [45,46,47]. At the telomere ends, DNA replication exhibits incomplete replication properties, which results in the loss of telomere sequence and telomere shortening, and then the telomerase enzyme can repair the telomere shortening caused by replication. This physiological telomere shortening is relatively tiny and constant and is mainly influenced by age. Because guanine has the lowest redox potential and the tandem repeat sequence of telomeres is rich in guanine, this suggests that telomere sequences are always at risk of oxidative damage [48]. Thus, massive oxidative base damage will accelerate the shortening of telomere sequences and may cause multiple diseases.

In recent years, it has been demonstrated that NEIL3 may preferentially remove oxidative damage from telomeric DNA, playing a large role in maintaining the stability of telomeres. The preceding section showed that NEIL3 can repair DNA damage in telomeric G-quadruplexes [44]. In addition, siRNA was used to specifically knock down the expression of NEIL3 in proliferating cells, which induced telomere defects and dysfunction, which in turn led to the formation of DNA bridges, aberrant mitoses and consequent cell death [34]. Significantly, the localization of NEIL3 at telomere sequences was enhanced and mainly concentrated in the S phase to the S/G2 late phase when DNA underwent oxidative stress damage during mitosis [34]. They further investigated the mechanism of NEIL3 recruitment linked to telomere sequences and how to initiate BER repair, which may be closely related to the disordered C-terminal domain (CTD) sequence of NEIL3 [34]. TRF1 and TRF2, the members of Shelterin proteins, bind to double-stranded telomeres, and their TRFH structural domains play an essential role in mediating protein–protein interactions [47]. The authors found that NEIL3 relied on its own CTD to interact with the TRFH structural domain of TRF1 and was recruited to telomeres, then NEIL3 bound to single-stranded DNA in a telomeric sequence-independent manner, as well as further increasing the glycosylase activity of NEIL3 [34]. In contrast, the telomeric DNA bound by NEIL3 was reduced by 5.2-fold after TRF1 knockdown, suggesting that TRF1 mediates the recruitment of NEIL3 to telomeres [34]. Then, when NEIL3 aggregated at telomeres, NEIL3 also relied on CTD sequences to activate APE1 activity and initiate the LP-BER pathway for participating in the repair of telomeric DNA damage [34]. This study illustrates that NEIL3 starts repairing telomere damage mainly during the S phase, which ensures chromosome division during mitosis. Later, researchers also demonstrated that NEIL3 initiated the BER pathway by recruiting APE1 and POLB to telomeric DNA damage sites primarily during mitosis, which in turn repaired telomere damage [49]. In vivo experiments revealed that *Apoe^-/-^Neil3^-/-^* mice had shortened telomeres and reduced telomerase activity in bone marrow cells compared to Apoe^-/-^ mice, as shown by reduced bone marrow production and lower leukocyte levels in peripheral blood [50]. This study suggests that NEIL3 has a role in protecting the length of telomeres in bone marrow cells in vivo, but the authors did not delve into the specific mechanisms.

### 4.5. NEIL3 Repairs DNA Interstrand Crosslinks (ICLs)

ICLs prevent replication and transcription by inhibiting DNA unwinding, which can cause severe chromatin distortion and is highly toxic to the organism [51,52]. Currently, drugs that can induce the formation of ICLs (such as platinum, nitrogen mustards, mitomycin c and psoralen, etc.) have been widely used in chemotherapy for cancers and leukemia, but this may also induce tumorigenesis [52,53,54]. Moreover, endogenous mutagens such as aldehydes or nitrites, which are metabolic by-products of cells in vivo, will continuously induce the accumulation of ICLs, and if the ICLs are not removed in time, apoptosis and cell death will be induced [52,54,55]. Therefore, this requires a well-established DNA–ICLs repair system. Studies have confirmed that two key repair pathways are immediately activated when ICLs cause complex fork arrest: the Fanconi anemia (FA) pathway and the NEIL3 pathway [56]. Critically, the E3 ubiquitin ligase TRAINP is an upstream regulator of the FA pathway and NEIL3 pathway activation, which ubiquitinates the CMG helicase; the short CMG ubiquitin chain can directly bind and recruit NEIL3. In contrast, the long CMG ubiquitin chain activates the FA pathway by unloading CMG [57].

The FA pathway is a critical signaling pathway in the repair system of DNA–ICLs, mainly including the FA core complex and FANCI/FANCD2 complex, and monoubiquitination modification of the FANCI/FANCD2 complex by the FA core complex is a crucial step for the FA pathway to start repairing ICLs [54]. Subsequently, activated FANCI/FANCD2 recruits multiple specific nucleic acid endonucleases (XPF-ERCC1, MUS81-EME1 and SLX1), which form incisions on both sides of the DNA and induce ICL unhooking; however, this process generates DSB, and then the HR pathway is required for further repair of DSB [58]. Once the FA gene is missing, the vulnerability of ICL will increase, resulting in the development FA genetic syndrome, with patients showing mainly low fertility, progressive bone marrow failure and cancer susceptibility [53].

It has been reported that NEIL3 knockout not only inhibited the proliferation of mouse embryonic fibroblasts but also increased the sensitivity of cells to oxidative toxicants (paraquat) and ICLs inducers (cisplatin) [32]. Additionally, NEIL3 played a more critical role in the repair process of psoralen-ICLs and abasic site ICLs (AP-ICLs) and did not produce DSB [59,60]. In a study concerning the repair pathway of ICLs in xenopus egg extracts, NEIL3 was found to specifically cleave two N-glycosyl bonds in AP-ICLs and psoralen-ICLs, allowing ICLs to unhook without forming DNA backbone cuts, which prevented the possibility of chromosomal rearrangements by not generating DSBs [60]. However, when N-glycosyl bond breaking was inhibited or NEIL3 was absent, the repair task of ICLs was performed by the FANCI/FANCD2-dependent incision pathway, which resulted in a significant increase in DSBs associated with oxidative or replicative stress, suggesting that the repair pathway of ICLs is selective [60]. Recently, studies in human cells further showed that the NEIL3 and FA pathways were non-epistatic in the repair of psoralen-ICLs [56]. Mechanistically, they found that the recruitment of NEIL3 to psoralen-ICLs was PARP-dependent and that the PARP inhibitor (olaparib) almost completely blocked the recruitment of NEIL3 to psoralen-ICLs [56]. In addition, the RUVBL1/2 complex participated in the FA pathway by maintaining the abundance of the FA core complex in cells [61]. At the same time, it also could bind physically to NEIL3 and act within the NEIL3 pathway [56]. Additionally, in psoralen-derived three- and four-stranded crosslinked DNA–DNA structures, human NEIL3 targeted cross-linked thymidine in ssDNA or dsDNA and produced two DNA cleavage products, one with an abasic site and another with a cross-linked dTMP to a thymine base, without producing single-stranded DNA breaks [59]. In summary, the NEIL3-mediated unhooking process of ICLs is the main repair pathway of AP-ICLs and psoralen-ICLs, while the FANCI/FANCD2-dependent incision pathway serves as a fallback mechanism to repair ICLs (Figure 2).

## 5. NEIL3 and Cancers

NEIL glycosylases can suppress cellular mutations and facilitate the maintenance of genomic DNA integrity. However, analysis of mRNA data regarding 13 human cancers showed that increased somatic mutation rates were associated with significantly reduced expression levels of NEIL1/2 and significantly increased expression levels of NEIL3, suggesting that NEIL3 overexpression instead promotes somatic mutation rates in cancer [62]. To investigate the reason, the authors found that high levels of NEIL3 expression were positively correlated with high levels of the mutation inducer (APOBEC3B) in 10 of the 13 cancers (76.9%), which may explain the increased number of somatic mutations induced by NEIL3 overexpression in cancer [62]. Moreover, *Neil1^-/-^*/*Nei23^-/-^* double knockout or *Neil1^-/-^*/*Neil2^-/-^*/*Neil3^-/-^* triple knockout mouse models did not show increased cancer susceptibility or spontaneous mutation frequency [63]. A previous study showed that the expression level of NEIL3 was significantly higher in 16 cancer tissues than in normal tissues, except for testicular and pancreatic cancer tissues [11]. Recent data showed that NEIL3 was overexpressed in various human cancers, including glioblastoma multiforme (GBM), breast cancer (BC), pancreatic adenocarcinoma (PADC), lung adenocarcinoma (LUAC), kidney renal clear cell carcinoma (KRCCC), kidney renal papillary cell carcinoma (KPCC) and low-grade glioma (LGG) [64]. Furthermore, NEIL3 overexpression was associated with poorer overall survival (OS) in patients with these cancers [64]. In contrast, data from patients with gastric and colorectal cancers suggested that NEIL3 overexpression was associated with better survival rates [64] (Table 1). This study suggests that high NEIL3 levels exert different prognostic effects in different cancers. The above studies suggest that NEIL3 is overexpressed in most cancers and may play a pro-cancer role as an oncogene. In addition, NEIL3 shows a role in predicting the progression, treatment responsiveness and survival of cancer patients, and it may be an independent prognostic indicator for certain cancers. This section will focus on the latest frontiers of NEIL3 in cancer.

### 5.1. NEIL3 and Hepatocellular Carcinoma (HCC)

Primary liver cancer is the sixth most common malignancy and the third leading cause of cancer death worldwide (8.3%), with approximately 900,000 new cases reported and 830,000 deaths per year, of which HCC is the most common histologic type [72]. In the last two years, researchers have gradually begun to discuss the relevance of NEIL3 in the development and prognosis of HCC. Previous high-throughput single nucleotide polymorphism array (SNP) analysis showed a high frequency of heterozygous loss of NEIL3 in HCC, suggesting that NEIL3 may rely on its DNA properties to act as a potential tumor suppressor gene in HCC [73]. In contrast, several studies revealed that NEIL3 mRNA and protein expression levels significantly overexpressed in HCC tissues or cell lines compared to paraneoplastic tissues or normal hepatocyte lines [49,65,66,74]. Regarding prognosis, high NEIL3 levels were positively associated with advanced TNM stage, tumor volume, tumor resistance and low survival time [65,74]. Subgroup analysis showed that patients with high NEIL3 levels had significantly shorter OS and DFS than those with low NEIL3 levels in HCC patients with advanced TNM stage, hypofractionated tumors, HBsAg-positive or cirrhotic liver [65] (Table 1). Zhao et al. found that when the DNA damage and repair pathway was activated, NEIL3 aggregated heavily to telomeres during mitosis while recruiting APE1 and POLB to telomeric DNA damage sites and then repairing telomeric DNA damage by initiating the BER pathway, thereby promoting proliferation and preventing senescence of HCC cells in vitro [49]. In addition, the knockdown of NEIL3 inhibited the proliferation and promoted the senescence of HCC cells but did not lead to apoptosis or necrosis [49]. Conversely, the migration and invasion ability of HCC cells was almost entirely rescued when the DNA repair-function-deficient NEIL3 mutant was reintroduced, suggesting that the power of HCC migration and invasion promoted by NEIL3 may not be closely linked to its DNA repair function, which differs from the study of Zhao et al. [74]. Recent work found that small nucleolar RNA host gene 3 (SNHG3) interacted with E2F1 and promoted its expression, which then promoted overexpression of NEIL3 by enhancing the binding ability of E2F1 and the NEIL3 promoter, ultimately promoting the malignant progression of HCC, and also suggested that SNHG3 and NEIL3 could be combined to assess the poor prognosis of HCC patients [74,75]. Mechanistically, NEIL3 directly interacted with the key EMT transcription factor TWIST1 and enhanced the phosphorylation level of TWIST1 through the BRAF/MEK/ERK axis, thereby suppressing the expression levels of E-cadherin and TIMP3 and promoting the expression levels of N-cadherin, MMP-9 and the drug resistance gene MDR-1, which in turn promoted EMT, invasion, lung metastasis and drug resistance [74]. In addition, NEIL3 partially induced the activation of the PI3K/Akt/mTOR signaling pathway in HCC, but the authors did not investigate in depth whether NEIL3 promotes HCC progression through activation of this pathway [66] (Figure 3). Given the effectiveness of inhibiting the PI3K/Akt/mTOR signaling pathway in treating HCC, this may be a direction worthy of further investigation. Recently, a Phase I clinical study on advanced HCC showed that patients showed good tolerance and potential therapeutic effects of a vaccine mixture containing NEIL3 peptide, which presents a new perspective for the treatment of HCC and deserves to be explored in depth [76]. The above studies illustrate that overexpression of NEIL3 levels can be an independent prognostic molecule in HCC patients.

### 5.2. NEIL3 and Non-Small Cell Lung Cancer (NSCLC)

NSCLC is the most common pathological subtype of lung cancer, including lung squamous cell carcinoma (LUSC) and lung adenocarcinoma (LUAD) [77]. NEIL3 has been reported to be significantly overexpressed in NSCLC tissues and cell lines, associated with more advanced clinical stage, larger tumors and poorer OS [67,68] (Table 1). Additionally, patients with NEIL3 overexpression showed a lower immunotherapeutic effect, while targeting NEIL3 significantly increased the sensitivity of patients to cisplatin and paclitaxel [68]. In a study on LUAD, NEIL3 overexpression was significantly negatively correlated with infiltration of B cells, CD4^+^ T cells and DCs in cancer tissues and significantly positively correlated with nTreg, iTreg, and exhausted T cells, suggesting that NEIL3 plays a crucial role in regulating immune infiltrating cells in LUAD and promotes the formation of a suppressive tumor microenvironment [67]. Gene set enrichment analysis (GSEA) further demonstrated that NEIL3 overexpression may contribute to poor outcomes in LUAD patients by activating downstream cell cycle pathways and P53 signaling pathways [67]. Similarly, the PI3K/AKT/mTOR pathway, G2/M checkpoint and E2F proteins were abnormally activated in patients with high NEIL3 expression, and we speculate that NEIL3 may significantly enhance the proliferation, invasion and migration of NSCLC cells by activating the PI3K/AKT/mTOR signaling pathway in part [68] (Figure 4). The above studies suggest that NEIL3 may become an independent predictor of prognosis in NSCLC patients, and inhibition of NEIL3 expression is expected to be an immune-related therapeutic target for NSCLC patients.

### 5.3. NEIL3 and Prostate Cancer

Castration-resistant prostate cancer (CRPC), neuroendocrine prostate cancer (NEPC) and chemotherapy-resistant prostate cancer are advanced prostate cancers; these three types of cancer have apparent resistance to androgen deprivation therapy (ADT), chemotherapy and radiotherapy and are the leading causes of death in patients with prostate cancer [78]. It was reported that among the three types of drug-resistant cancer, the overexpression of NEIL3 was associated with a higher T/N stage and Gleason score, but it was positive with a good outcome for patients [69,70]. Interestingly, NEIL3 was barely expressed in the tissues or cells of CRPC, NEPC and chemoresistant-prostate cancer, suggesting that NEIL3 may be associated with developing prostate cancer resistance [69]. In the cancer cell models that received radiotherapy or chemotherapy (nzarotamide, docetaxel and cisplatin), NEIL3 knockdown significantly reduced the sensitivity of prostate cancer cells to radiotherapy or chemotherapy but did not affect ADT resistance [69,70] (Table 1). Mechanistically, the absence of NEIL3 significantly triggered the activation of phosphorylation of the ATR-CHK1 axis, which mitigated DNA damage by initiating the DNA damage response (DDR) pathway, thereby stabilizing replication forks and delaying cell cycle progression, ultimately conferring resistance to radiotherapy or chemotherapy in prostate cancer cells [69,70] (Figure 4). Therefore, the therapeutic modality of radiotherapy or chemotherapy combined with NEIL3 inducers may hold some promise in the therapeutic study of prostate cancer.

### 5.4. NEIL3 and GBM

Astrocytoma is a common malignant tumor of the brain [79]. GBM is the most malignant type of astrocytoma with strong mitogenic activity, and its rapid progression and high drug resistance result in low patient survival rates [79,80]. Gene polymorphism analysis indicated that the rs12645561 type in NEIL3 may be associated with the progression of GBM in the Chinese Han population [81]. The expression level of NEIL3 was one of the two most significantly increased genes in GBM patient samples, possibly associated with a high capacity for invasion; its overexpression was independently associated with a higher risk of death in GBM patients [71]. Furthermore, in NEIL3 knockout GBM cell lines, we observed the G2/M phase cell cycle arrest and a significant increase with replication-associated DSB; radiation treatment based on NEIL3 knockout further increased the mortality of tumor cells [71]. This study suggests that NEIL3 deficiency inhibits the mitotic activity of GBM cells and increases the sensitivity of GBM cells to radiotherapy (Table 1). In vitro experiments also revealed that NEIL3 deficiency increased the sensitivity of GBM cells to ATR inhibitors and PARP1 inhibitors (olaparib) [36] (Figure 4). This study suggests that for patients with NEIL3-deficient tumors, the combination of ATR inhibitors with olaparib may be useful in treating glioblastoma patients, which has significant clinical implications.

### 5.5. The Role of circNEIL3 in Cancers

Circular RNAs (circRNAs) are a class of non-coding RNAs that are abundant, conserved and stable in living organisms, and it has more significant potential in research of cancer therapeutic targets [82,83]. Many studies have shown that circRNAs eliminate the repressive effects of microRNAs (miRNAs) against target genes mainly by acting as miRNA sponges [84]. More importantly, circRNA sufficiently expresses in body fluids such as blood, urine and saliva, and it can be used for diagnostic and therapeutic studies of cancers [85]. circNEIL3 derives from the cyclization of exons 8 and 9 of NEIL3 gene and is stably present in the cytoplasm; overexpression or knockdown of circNEIL3 does not affect the transcription and expression of NEIL3 [86,87,88]. Recent studies have shown that circNEIL3 may be involved as an oncogene in the development of several cancers, such as LUAD, cervical cancer, GBM and pancreatic cancer. This section will focus on the possible mechanisms by which circNEIL3 promotes the development of the above cancers.

circNEIL3 significantly overexpressed in HCC tissues or cell lines, and specific knockdown significantly inhibited the malignant behavior of HCC cells and tumor growth in mice, suggesting that circNEIL3 is an oncogenic gene in HCC [88]. Laminin subunit γ 1 (LAMC1) has been shown to promote the proliferation and migration of HCC cells as an oncogene and is associated with poor prognosis in HCC patients [89]. In addition, the data suggested that miR-3150b-3p could directly bind to the 3’UTR of LAMC1 and inhibit the expression of LAMC1 to exert oncogenic effects [88]. In HCC, circNEIL3 targeted binding sites of miR-3150b-3p and repressed miR-3150b-3p expression, thereby indirectly promoting LAMC1 expression and ultimately promoting the progression of HCC [88] (Figure 5).

CircNEIL3 was the most significantly downregulated circRNA in LUAD treated with 0, 2 or 4 Gy radiotherapy, while circNEIL3 knockdown enhanced the sensitivity of xenograft tumors to radiotherapy [87]. Further, circNEIL3 overexpression significantly inhibited the occurrence of radiotherapy-induced LUAD cell pyroptosis and promoted cancer cell growth, while the knockdown of the PIF1 decapping enzyme reversed the effect of circNEIL3 overexpression [87]. Further mechanisms showed that circNEIL3 eliminated the inhibitory effect of miR-1184 on PIF1 by acting as miRNA sponges, and elevated PIF1 inhibited the activation of the AIM2 inflammasome by reducing DNA damage, ultimately promoting LUAD cell pyroptosis and enhancing the sensitivity of LUAD cells to radiotherapy [87] (Figure 5). This finding suggests that the circNEIL3/miR-1184/PIF1 axis inhibits radiation-induced cellular pyroptosis by regulating the DNA damage pathway.

Cervical cancer is the second most common malignancy in the world for women and the leading cause of death in women with cancer [72]. A study found that circNEIL3 was highly expressed in cervical cancer tissues and cells [90]. Mechanistically, circNEIL3 indirectly upregulated the expression of oncogene KLF12 by suppressing the expression of oncogene miR-137, which ultimately enhanced the proliferation of cervical cancer cells, while inhibiting circNEIL3 produced the opposite result [90] (Figure 5). This finding suggests that circNEIL3 promotes cervical cancer progression by activating the circNEIL3/miR-137/KLF12 axis.

CircNEIL3 was also significantly upregulated in glioma tissue compared to normal brain tissue, and its expression level was positively correlated with tumor grade, which may be a predictor of poor prognosis in glioma patients [91]. This study also revealed that EWS RNA binding protein 1 (EWSR1) promoted the biosynthesis of circNEIL3 by binding to the corresponding sequence of NEIL3 pre-mRNA, which was considered one of the cyclization mechanisms of circNEIL3 [91]. Insulin-like growth factor 2 mRNA binding protein 3 (IGF2BP3) is a known oncogenic protein molecule associated with the malignant progression of gliomas. HECTD4 is a protein member of the E3 ubiquitin ligase HECD family, which can degrade IGF2BP3 via the ubiquitin–proteasome pathway. We found that circNEIL3 inhibited the binding of IGF2BP3 to HECTD4 in vitro, which suppressed the ubiquitination degradation of IGF2BP3 by HECTD4, thus promoting the expression of IGF2BP3 protein and its downstream oncogenic proteins such as CDK4/6, CD44 and c-MYC, and ultimately promoting the proliferation, invasion and migration of GBM cells [91]. Tumor-associated macrophages (TAMs) are the primary immune cells infiltrating glioma tissue and are associated with the development of immunosuppression. We found that circNEIL3 could also promote the secretion of macrophage chemokines (CCL2 and LOX) by activating YAP1 signaling, which induced massive aggregation of TAMs in the tumor microenvironment [91]. In addition, hnRNPA2B1 could package circNEIL3 into exosomes and deliver them to infiltrating TAMs; circNEIL3 promoted the conversion of TAMs to an immunosuppressive phenotype by stabilizing IGF2BP3, which ultimately promoted glioma progression [91] (Figure 5).

RNA editing is involved in cancer development. The adenosine deaminases acting on the RNA (ADAR) family of enzymes are critical in RNA editing, including ADAR1 and ADAR2. Among them, ADAR1 is considered a pro-oncogene and plays a role in facilitating adenosine-to-inosine (A-to-I) editing in dsRNA. Studies have confirmed that endogenous ADAR1 can directly bind to the mRNA of glioma-associated oncogene 1 (GLI1) and then promote its transcriptional activity by inducing RNA editing of GLI1, and the edited GLI1 is involved in a variety of cancers. In vivo and in vitro experiments on pancreatic ductal adenocarcinoma confirmed that circNEIL3 could promote proliferation, migration and invasion of PDAC, and circNEIL3 overexpression was positively associated with the number of metastatic nodules in the lung and liver [86]. In contrast, circNEIL3 knockdown had the opposite effect [86]. Further investigation revealed that circNEIL3 acted as a miR-432-5p sponge and repressed its expression, which led to the activation of the ADAR1/GLI1 axis [86]. Then, GLI1 promoted cell cycle progression by promoting the expression of Cyclin D1/CDK4/CDK6 complex and induced EMT by activating the Snail pathway [86]. Interestingly, ADAR1 could also induce the cyclization of circNEIL3 via negative feedback [86] (Figure 5). Meanwhile, clinical data showed that high expression of circNEIL3, low expression of miR-432-5p and high expression of ADAR1 predicted poorer OS in PDAC patients, with circNEIL3 and ADAR1 expression levels possibly being independent prognostic factors in PDAC patients [86]. 

## 6. Related Studies of NEIL3 and Other Diseases

### 6.1. NEIL3 and Cardiovascular Diseases (Atherosclerosis, Myocardial Infarction, Myocardial Rupture)

Atherosclerosis remains the leading cause of death in elderly patients, with common risk factors including smoking, dyslipidemia, diabetes, inflammation and oxidative stress, which can promote the sustained release of ROS or reactive nitrogen species (RNS) within plaques [92]. Studies have demonstrated that the development and progression of atherosclerotic plaques are associated with the accumulation of DNA damage, such as nuclear or mitochondrial DNA damage in vascular smooth muscle cells (VSMC), macrophages and endothelial cells [93]. A study found that human atherosclerotic plaques had more DSBs in VSMC DNA, and this led to restricted growth and increased apoptosis of VSMC, which decreased the fibrous cap’s relative area on the plaque surface and promoted plaque instability [94]. Studies suggested that deficiency of the BER pathway may accelerate atherosclerosis formation, and NEIL3 was shown to have increased expression in human atheromatous plaques and may play a role in inhibiting atherosclerosis formation, but whether it relies on DNA repair capacity is unclear. Reportedly, when male *Apoe* gene-deficient (*Apoe^-/-^*) mice were exposed to a high-fat diet, NEIL3 gene deficiency (*Neil3^-/-^*) accelerated atherosclerosis formation, and interestingly, genomic integrity and accumulation of oxidative DNA damage in plaques were not significantly altered, suggesting that *Neil3^-/-^* may accelerate atherosclerosis formation through other mechanisms [95]. It was demonstrated in 2016 that the susceptibility of *Apoe^-/-^Neil3^-/-^* mice to atherosclerosis might be related to the following two factors [96]: (1) *Apoe^-/-^Neil3^-/-^* affected lipid metabolism, resulting in significant elevation of hepatic TG, total fatty acid and monounsaturated fatty acid levels; (2) *Apoe^-/-^Neil3^-/-^* induced macrophage infiltration in plaques and reduced cholesterol efflux capacity of macrophages by downregulating ABC transporter protein, leading to uneven lipid distribution [96]. In addition, in Apoe^-/-^ mice, *Neil3^-/-^*-promoted atherosclerotic plaque formation may also be associated with VSMC proliferation, lipid accumulation, and transformation into a pro-atherosclerotic cell phenotype, which may be dependent on the activation of AKT signaling [96]. A recent study reported that *Apoe^-/-^Neil3^-/-^* might also promote atherogenesis by altering the composition of the intestinal microbiota (Faecalibaculum was decreased and Lactobacillales was increased), increasing intestinal permeability and plasma lipopolysaccharide levels to induce changes in the food metabolic profile of mice [97]. Moreover, a nested case–control study revealed that genetic variation of NEIL3 rs12645561 SNP TT genotype was associated with an increased risk of myocardial infarction [98]. Further studies confirmed that NEIL3 was expressed at elevated levels in the myocardial tissue of patients with heart failure and MI, especially in the fibroblast population, as NEIL3 could finely regulate fibroblast proliferation through DNA methylation, thereby fine-tuning matrix degradation and fibrillogenesis within the myocardium and ultimately forming stable scarring, which may help prevent cardiac rupture [99]. However, in *Neil3^-/-^*-deficient mice, there was a significantly increased risk of a myocardial rupture occurring after myocardial infarction due to dysregulated proliferation and differentiation of fibroblasts caused by *Neil3^-/-^* deficiency, resulting in increased early mortality [99]. The above studies illustrate that appropriate NEIL3 expression levels help prevent the development of cardiovascular diseases such as atherosclerosis, myocardial infarction, and myocardial rupture.

### 6.2. NEIL3 and Neurological Disorders

The brain is an organ with active cellular metabolism; along with releasing highly oxidizing substances and free radicals, nerve cells are constantly exposed to oxidative stress stimulation, which requires a powerful oxidative DNA repair system inside the brain [100]. In mouse brain tissue, NEIL1, NEIL2, OGG1 and NTH1 showed widespread expression at all ages; NEIL3 showed a discrete expression pattern only in brain regions containing stem cell populations in neonatal mice; the expression level of NEIL3 decreased with age [16]. There has been sufficient evidence that NEIL3 is closely associated with the induction and maintenance of neurogenesis, particularly in the maintenance of adult hippocampal neurogenesis and cognitive performance, suggesting a link between NEIL3 and diseases such as neurodegenerative disorders and aging. In cells with knockdown of NEIL3, embryonic neural stem cells had reduced differentiation potential and proliferation rate and showed signs of premature senescence [101]. An in-depth study revealed that NEIL3 might stabilize the proliferation and differentiation of neural stem/progenitor cells (NSPCs), neuronal progenitor cells, astrocytes, and microglia by repairing Sp and Gh damage in ssDNA, thus promoting neurogenesis and resisting neurological damage caused by hypoxia [102,103], while *Neil3^-/-^* mice partially lost the function of repairing Sp and Gh damage in ssDNA, resulting in impaired hippocampal neurotransmission systems and constitutive changes in synapses, exhibiting learning and memory deficits and reduced anxiety-like behavior [103]. On the other hand, *Neil3^-/-^* neonatal mice exhibited delayed maturation of hippocampal CA1 neurons, impaired spatial stability of hippocampal CA1 positional cells, and affected spatial cognition in the hippocampus [104]. Additionally, prion disease is a severe neurodegenerative disease, and NEIL3 seemed to protect brain tissue from oxidative damage by inducing neurogenesis, which could prevent prion disease in the clinical stage [105]. Ginsenoside Rd (GSRd), a kind of extract of the Chinese traditional medicine ginseng, reduced mitochondrial and nuclear DNA damage by possibly upregulating the expression of NEIL1 and NEIL3, thereby inhibiting apoptosis and restoring neurological function in the brain, with better neuroprotective function in an acute ischemic stroke rat model [106].

## 7. Conclusions and Outlook

The current state of research indicates that NEIL3 is one of the critical glycosylases in the repair process of DNA base damage and plays an essential role in maintaining genome stability. Unlike other DNA glycosylases, NEIL3 has its structural properties and excels in the repair of bases, including favoring base damage at ssDNA, G-quadruplex, ICLs, replication forks and telomere sites, thus promoting cell cycle progression and cell proliferation. A series of studies have confirmed that *Neil3^-/-^* would probably lead to decreased bone marrow hematopoietic capacity, autoimmune defects, cardiovascular disease (atherosclerosis, myocardial infarction, myocardial rupture) and neurological disorders [50,96,98,99,102,107]. On the other hand, unlike NEIL1/2, overexpression of NEIL3, in turn, significantly increased the frequency of mutations in somatic cells; public database analysis also showed that NEIL3 was significantly overexpressed in a variety of cancer tissues or cancer cells, suggesting that NEIL3 may be a carcinogenic gene. Therefore, either excessively high or low levels of NEIL3 expression may contribute to the development of diseases, and its transcription and translation need to be tightly regulated. Recent studies have found that high levels of NEIL3 and circNEIL3 could promote carcinogenesis, metastasis and drug resistance in several ways, including HCC, NSCLC, GBM, cervical cancer and PDAC, and were strongly associated with poor prognosis of these cancers. Interestingly, NEIL3, although also overexpressed in gastric, colorectal and prostate cancers and promoting their progression, was associated with a good prognosis. Therefore, NEIL3 overexpression levels exert different prognostic effects in different cancers, and further comprehensive evaluation of the magnitude of NEIL3 value as an independent prognostic biomarker in cancer is still needed. In terms of therapeutic studies, vaccine mixtures containing NEIL3 peptides have shown new therapeutic directions in HCC treatment studies [76], and NEIL3 knockdown increased the sensitivity of GBM cells to radiotherapy, ATR inhibitors and PARP1 inhibitors [35], suggesting that the combination approach of targeted inhibition of NEIL3 with ATR/PARP1 inhibitors may become a new therapeutic target for some cancers. However, research on NEIL3 and circNEIL3 with cancers is still lacking, especially the number of studies related to the mechanism, prognosis and treatment is scarce and needs further exploration.

## Figures and Tables

**Figure 1 cancers-14-05722-f001:**
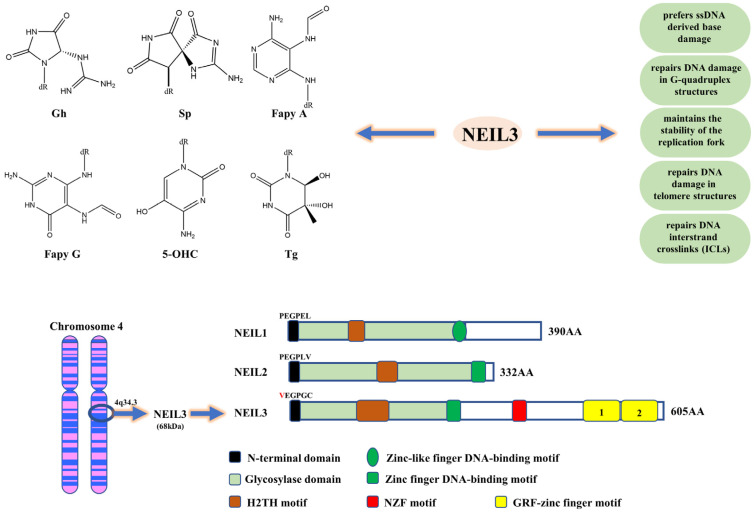
The structural and functional characteristics of NEIL3. Gh, Sp, FapyG, FapyA, 5-OHC and Tg are the main DNA base damage substrates of NEIL3. NEIL3 prefers base damage in ssDNA or G-quadruplex, maintains replication fork stability, and preferentially repairs telomeres and ICLs. The human NEIL3 gene is located on chromosome 4q34.3. The human NEIL3 protein is composed of an N-terminal domain, an H2TH motif, a zinc finger DNA-binding motif, a ZNF motif and a duplicated GRF-ZF motif. In the N-terminal domain of NEIL3, valine replaces the central catalytic role of proline.

**Figure 2 cancers-14-05722-f002:**
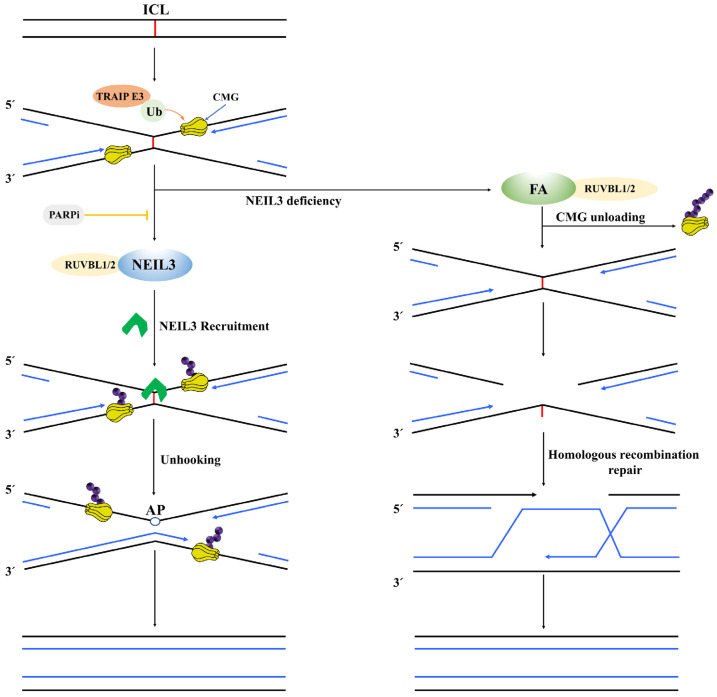
A schematic diagram of the NEIL3 pathway and FA pathway synergistically repairing ICL. First, the E3 ubiquitin ligase TRAINP ubiquitinates the CMG helicase; the short CMG ubiquitin chain can directly bind and recruit NEIL3. NEIL3 specifically cleaves two N-glycosyl bonds in ICLs, allowing ICLs to unhook without generating DSBs. When NEIL3 is deficient, the long CMG ubiquitin chain activates the FA pathway by unloading CMG; however, this process generates DSB, and then the HR pathway is required for further repair of DSB.

**Figure 3 cancers-14-05722-f003:**
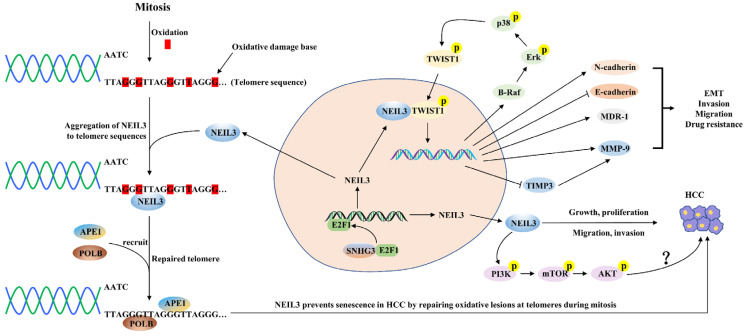
A schematic diagram of the mechanisms by which NEIL3 promotes HCC progression. SNHG3 interacts with E2F1 and enhances the binding ability of the E2F1 and NEIL3 promoter, thus promoting NEIL3 overexpression. NEIL3 promotes the malignant progression of HCC through the following mechanisms: (1) repairing telomere damage in HCC; (2) promoting EMT, invasion, lung metastasis and drug resistance by enhancing the phosphorylation level of TWIST1; and (3) promoting HCC cell proliferation possibly by activating the PI3K/AKT/mTOR axis.

**Figure 4 cancers-14-05722-f004:**
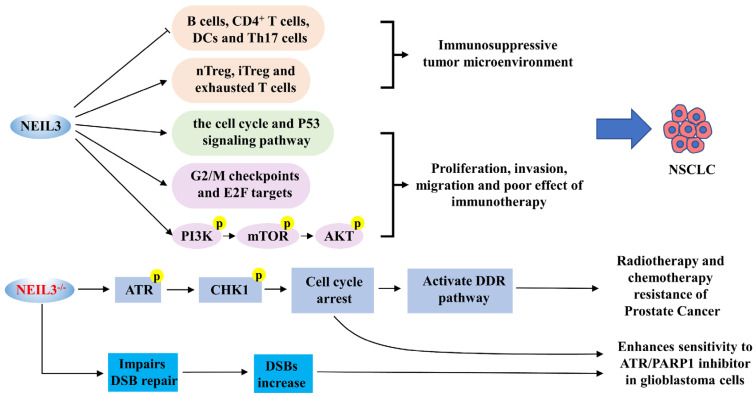
A schematic diagram of NEIL3 and the progression of NSCLC, prostate cancer and GBM. NEIL3 promotes NSCLC proliferation, invasion and migration by inducing the formation of an immunosuppressive tumor microenvironment, accelerating the cell cycle, and activating P53 and PI3K/AKT/mTOR pathways. The deficiency of NEIL3 significantly activates the ATR/CHK1 axis, which attenuates DNA damage by initiating the DDR pathway, ultimately conferring resistance to radiation or chemotherapy in prostate cancer cells. Nevertheless, NEIL3 deficiency increases the sensitivity of GBM cells to ATR inhibitors and PARP1 inhibitors.

**Figure 5 cancers-14-05722-f005:**
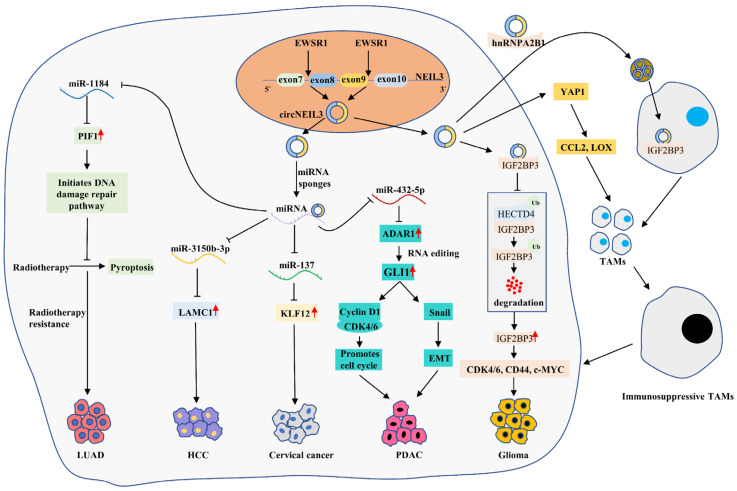
A schematic diagram of circNEIL3 promoting the malignant progression of LUAD, HCC, cervical cancer, GBM and PDAC. EWSR1 promotes the cyclization of circNEIL3 by binding to exons 8 and 9 of NEIL3 pre-mRNA. circNEIL3 promotes HCC progression by regulating the circNEIL3/miR-3150b-3p/LAMC1 axis. circNEIL3 significantly inhibits the generation of radiotherapy-induced LUAD cell pyroptosis and promotes carcinoma cell growth by regulating the circNEIL3/miR-1184/PIF1 axis. circNEIL3 promotes cervical cancer cell proliferation by regulating the circNEIL3/ miR-137/ KLF12 axis. circNEIL3 promotes glioma proliferation, invasion and migration by regulating the circNEIL3/HECTD4/IGF2BP3 axis, and it also induces the massive aggregation of immunosuppressive TAMs in the tumor microenvironment by activating the YAP1 and exosome pathways. circNEIL3 promotes cell cycle progression and EMT through regulation of circNEIL3/miR-432-5p/ADAR1/GLI1 axis, ultimately promoting PDAC proliferation, migration and invasion.

**Table 1 cancers-14-05722-t001:** Prognostic value of NEIL3 in related cancers.

Cancer Type	Prognostic Value	References
HCC	Patients with high NEIL3 expression had worse OS and disease-free survival (DFS) than those with low expression.	[65]
HCC	NEIL3 overexpression was negatively correlated with survival and progression-free survival in HCC patients and positively correlated with the progression of the TNM stage.	[49]
HCC	NEIL3 overexpression correlated with tumor grade and lower 10-year survival rates but not with age, sex, race, or weight.	[66]
LUAD	NEIL3 overexpression significantly reduced OS and RFS in LUAD patients, positively associated with advanced stage, tumor volume, and poorer OS.	[67,68]
Prostate cancers	NEIL3 was barely expressed in tissues or cells of CRPC, NEPC and chemoresistant prostate cancer; high levels of NEIL3 were associated with higher T/N stage and Gleason score but were positively associated with good prognosis in prostate cancer patients.	[69,70]
GBM	Overexpression of NEIL3 was independently associated with poor patient prognosis.	[71]
Multiple cancers	High NEIL3 expression levels predicted worse prognosis and higher risk of death in patients with GBM, TNBC, PADC, LUAC, KRCCC, KPCC and LGG, but instead were good prognostic indicators for patients with colorectal and gastric cancers.	[64]

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
