# Peer review of "Biological Functions of the DNA Glycosylase NEIL3 and Its Role in Disease Progression Including Cancer"

_cancers, 2022, doi:10.3390/cancers14235722_

Round 1

Reviewer 1 Report

To Chen et al, Biological functions of the DNA glycosylase NEIL3 and its role cancers progression,  Cancers

The authors of this review summarize the published data on NEIL3 in a thorough and ambitious way. They go into the substrate specificity and repair mechanisms of NEIL3 as well as its role in tumor progression of specific cancer types. 

I have a major concern about the quality of the language which must be improved to increase the readability and clarity of the review. The abstract and introduction were well written but some of the later parts of the text need to be improved. I suggest that a native English speaker should proofread the manuscript to improve the language quality.

I have several points of concern that need to be addressed:

-The section 3.2 NEIL3 maintains the stability of the replication fork was very unclear and hard to read. Many of the sentences were too long and complicated which made it hard to follow the reasoning. The language was poor, e.g.the sentences on line 159 “Someone found…” and line 165: “Study suggested…” and the whole section need major revision. I also found some references missing on line 162 at which only ref 12 is referred to but ref 48 is also showing this.

- In figure 1, it would be informative to include information on the function of each protein domain of NEIL3 and their interaction partners.

- On line 195, please rephrase the expression “biological enzymes” to a more general term for example “DNA repair systems”.

- On line 195, please specify which glycosylases that are referred to in “none of the glycosylases”. 

- The sentence on line 229 is unclear and needs rephrasing. What does “confirmed the characterization” mean? And why is “above” added to the end of the sentence?

-The authors use “telomeric sequences” throughout the text but it would be more correct to use “telomeres” instead.

- In section 3.4, the authors do not include a data from ref 65 which show that the repair of telomeric DNA damage occurs in mitosis.

- Line 241, please remove the word ‘intrinsic’ which is redundant.

- The sentence on line 361 beginning with “In addition” states that reference 65 (Zhao et al) shows that NEIL3 knockdown inhibit migration and invasion and induce apoptosis which it does not. The study shows that NEIL3 knockdown inhibit proliferation and promote senescence. Please correct this mistake.

- There were several sentences starting with ‘Authors’ which should be rephrased to improve the language quality.

Reviewer 2 Report

The article by Chen et al. provides an excellent review of NEIL3 function and its role in pathological diseases, most notable cancer.  I feel that this article will be of great interest to scientists studying DNA repair, genomic fidelity, and cancer.  However, I have a few suggestions that may improve the overall quality of this very good manuscript.

Suggestion 1: I would include a figure that provides the chemical structures of the DNA lesions that are substrates for NEIL3.

Suggestion 2: It would be nice to have a figure that compares the structure of NEIL3 with other DNA glycosylases.  In fact, I must insist that the authors include a figure that illustrates similarities and differences in the structure and/or structural organization of NEIL3 with other DNA glycosylaases.

Suggestion 3: Table 1 must be reformatted. In its current form, the table reads more like a complete paragraph.  It would be far more valuable to simply highlight some of the key features from the studies described in this table.

Suggestion 4: Section 5 of the article discusses the role of NEIL3 in pathological conditions that are distinct from cancer.  This contradicts the title of the article which states only cancer.  Based on this, I recommend following. Remove Section 5 so as to focus only on cancer or change the title to reflect the role of NEIL3 in different pathological conditions. 

Suggestion 5:  While they article is rather easy to follow, there are several places in the text that contain awkward or inaccurate statements that detract from the quality of the manuscript. For example, on line 154, the authors use the term "unstranding".  This is not acceptable. There are numerous other instances throughout the manuscript.  I would strongly recommend that the authors carefully review the document and correct inappropriate and awkward phrases.   
